# Circular RNA biogenesis can proceed through an exon-containing lariat precursor

**Steven P Barrett[1]\*, Peter L Wang[1,2], Julia Salzman[1,2]\***

[1]Department of Biochemistry, Stanford University School of Medicine, Stanford, United States; [2]Stanford Cancer Institute, Stanford University School of Medicine, Stanford, United States

**Abstract** Pervasive expression of circular RNA is a recently discovered feature of eukaryotic gene expression programs, yet its function remains largely unknown. The presumed biogenesis of these RNAs involves a non-canonical 'backsplicing' event. Recent studies in mammalian cell culture posit that backsplicing is facilitated by inverted repeats flanking the circularized exon(s). Although such sequence elements are common in mammals, they are rare in lower eukaryotes, making current models insufficient to describe circularization. Through systematic splice site mutagenesis and the identification of splicing intermediates, we show that circular RNA in *Schizosaccharomyces pombe* is generated through an exon-containing lariat precursor. Furthermore, we have performed high-throughput and comprehensive mutagenesis of a circle-forming exon, which enabled us to discover a systematic effect of exon length on RNA circularization. Our results uncover a mechanism for circular RNA biogenesis that may account for circularization in genes that lack noticeable flanking intronic secondary structure.

\*For correspondence: sbarret1@
stanford.edu (SPB); julia.
salzman@stanford.edu (JS)

**Competing interests:** The authors declare that no competing interests exist.

**Reviewing editor**: Douglas L Black, Howard Hughes Medical Institute, University of California, Los Angeles, United States

## Introduction

Pervasive expression of circular RNA is a recently discovered feature of eukaryotic gene expression; these circular isoforms are ubiquitously expressed in humans and a number of highly diverged eukaryotic organisms and can exceed the level of the cognate mRNA (*Salzman et al., 2012*; *Memczak et al., 2013*; *Salzman et al., 2013*; *Wang et al., 2014*). Rare examples of circular RNA expression had been known to exist for decades, but with a handful of exceptions, only known to exist at levels suggesting they were transcriptional noise or in vitro artifacts (*Nigro et al., 1991*; *Cocquerelle et al., 1992*; *Zaphiropoulos, 1996*). The basic mechanism responsible for the biogenesis of circular RNA involves a 'backsplicing' reaction in which a branch point upstream of a circularized exon attacks a downstream splice donor, positioning the 3′ end of the exon to attack its own 5′ end in the second step (*Figure 1*) (*Braun et al., 1996*; *Pasman et al., 1996*; *Schindewolf et al., 1996*).

Until recently, one of the only known examples of an abundant circular RNA was discovered in the mouse sex-determining gene, SRY (*Capel et al., 1993*). This exon is flanked by long (~15 kb) inverted sequence whose complementarity is required for circularization (*Dubin et al., 1995*). Early in vitro studies also demonstrated that complementary sequences flanking an exon can promote circularization, though they did not appear necessary (*Pasman et al., 1996*). Modeled after work on SRY, recent studies in mammalian cell culture have found similar results in human genes (*Liang and Wilusz, 2014*; *Zhang et al., 2014b*), and additional genome-wide computational sequence analysis suggests this complementary sequence-mediated mechanism may be widespread and associated with Alu elements (*Jeck et al., 2013*; *Zhang et al., 2014b*; *Ivanov et al., 2015*).

**eLife digest** DNA contains the instructions to make proteins. These instructions are first copied into a molecule of RNA, which often has sections that code for protein (called exons) interrupted by non-coding sections (called introns). A process called splicing removes the introns and joins the exons to form a mature RNA molecule that is then translated to make proteins.

The standard splicing process joins the first exon to the second, and the second to the third, and so on to produce a liner RNA molecule. Occasionally, one or more of the exons can be skipped in a process known as 'alternative splicing'. In recent years, scientists have discovered that another type of alternative splicing, which creates circular RNA molecules, is common in animal cells. It was suggested that circular RNAs might form when the splicing machinery joins the two ends of the same exon via a process referred to as 'backsplicing'.

Standard splicing first converts an intron into a looped structure called a 'lariat' (which resembles a lasso); the lariat is then removed by the splicing machinery and destroyed. When exons are skipped during alternative splicing, a large lariat containing the skipped exon is produced. Now, Barrett et al. have used yeast cells to investigate how circular RNAs are formed. Yeast is a good model to study this process because it can be manipulated easily in the laboratory and expresses circular RNAs. The experiments show that lariat structures containing exons (as would be expected from exon-skipping) are a common intermediate step before the production of a circular RNA. However, some exon-containing lariats do not go on to form circular RNAs, suggesting that additional factors are important for the production of circular RNAs.

Barrett et al. next investigated the reasons why certain exons formed circular RNAs while others did not and revealed that longer exons formed circular RNAs more readily than shorter exons. Further work is now required to understand the molecular basis of this result and identify other factors that contribute to the formation of circular RNAs.

Although widespread in mammalian genomes, repeat sequences are far less common in the genomes of simple eukaryotes, indicating that an alternative mechanism may be at play in these organisms and in human genes that lack inverted repeats. In addition, a recently published study in *Drosophila* fails to find a remarkable relationship between secondary structure and circular RNA biogenesis, noting that there are no noticeable sequence motifs in the introns flanking circularized exons besides canonical splice site sequences (*Westholm et al., 2014*).

An alternative mechanism that has long been proposed in the literature involves an exon-containing lariat, which can serve as a precursor to the circularized exon (*Figure 1*, right) (*Zaphiropoulos, 1996*; *Surono et al., 1999*; *Burd et al., 2010*; *Jeck et al., 2013*; *Jeck and Sharpless, 2014*). An early study noted a correlation between exon-skipping events and circular RNA isoforms in the human cytochrome P450 2C18 gene, describing four alternative circularization events and their corresponding exon-skipped transcripts (*Zaphiropoulos, 1996*); a similar correspondence was noted in the dystrophin gene (*Surono et al., 1999*). Recently, a study in human cell culture posits a correlation between exon skipping and circular RNA biogenesis in a model gene, but treats the exon-skipping event as a byproduct rather than a precursor of the backsplicing event, regarding inverted sequences as the determinant of circularization (*Zhang et al., 2014b*). Indeed, the only current evidence for this model is based on correlation between exon skipping and circular RNA production in a handful of genes. No biochemical evidence of the activity of this pathway has been provided, and no global relationship exists between exon skipping and circular RNA biogenesis.

Due to the large size of human genes, cell culture studies have been unable to ectopically express entire circle-forming genes on a plasmid and therefore have not been able to completely recapitulate splicing from the endogenous locus. Instead, expressed minigenes containing circle-forming exons and immediately flanking sequence have been used as surrogates (*Ashwal-Fluss et al., 2014*; *Liang and Wilusz, 2014*; *Zhang et al., 2014b*; *Starke et al., 2015*). These expression vectors may produce results that are due to characteristics of transcription in the minigene and, in some cases, generate large amounts of off-target splice products. The ability to recapitulate the

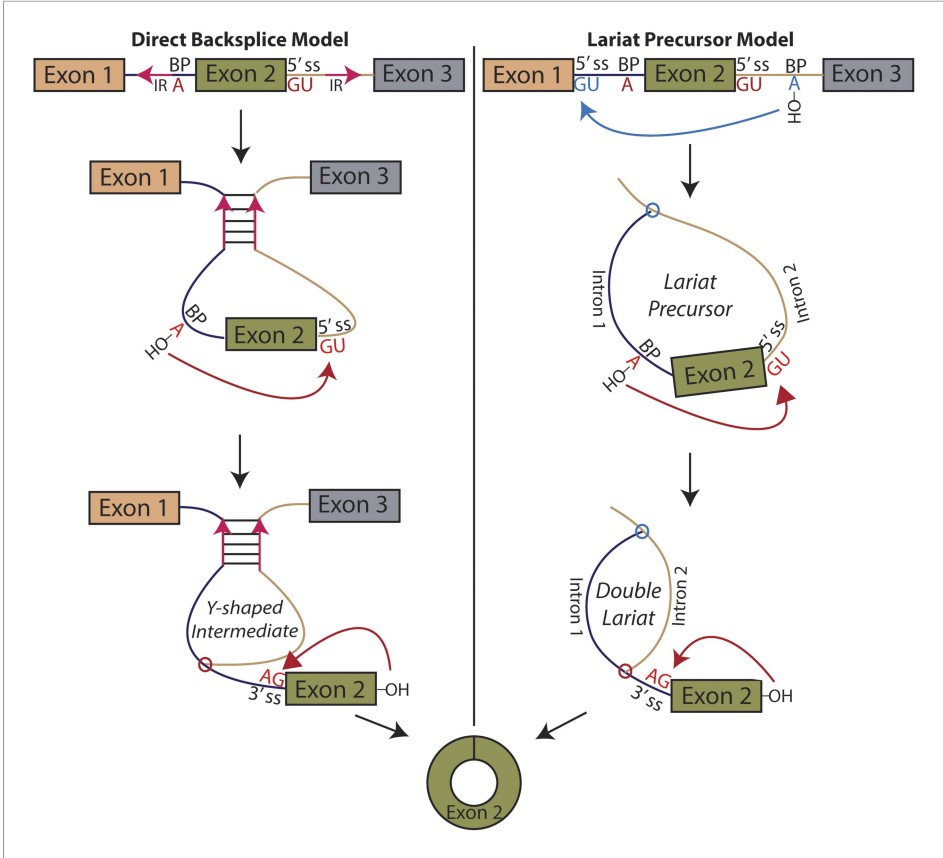

**Figure 1**. Models for the production of circular RNA. (*Left*) Complementary sequences facilitate the biogenesis of circular RNA. (*Right*) Circular RNA biogenesis proceeds through a lariat precursor.

The following figure supplements are available for figure 1:

**Figure supplement 1**. *mrps16* pre-mRNA predicted pair probabilities.

**Figure supplement 2**. *mrps16* pre-mRNA predicted MFE structure.

splicing of an endogenous locus is a powerful tool for studying the mechanism of circular RNA production. For this reason, we turned to *Schizosaccharomyces pombe* as our model system for studying this process. With a simple and small gene structure and relatively high expression of some circular RNAs, *S. pombe* is an ideal system for studying circular RNA biogenesis. Because the second exon of *mrps16* is the most highly expressed circular RNA in this organism (*Wang et al., 2014*), we have adopted *mrps16* as our model RNA for studying circular RNA production in the present work.

Utilizing the lariat debranching mutant (*dbr1Δ*), we provide direct evidence of an exon-containing lariat precursor and the double lariat resulting from lariat re-splicing (*Figure 1*, right). Furthermore, by expressing *mrps16* on a plasmid, we completely recapitulate the quantitative relationship between circular and linear RNA expression in this gene, and through systematic splice site deletions of this plasmid, we demonstrate the necessity of a lariat precursor for circular RNA biogenesis. However, lariat production is not sufficient for circularization, as demonstrated by the existence of several exon-containing lariats that do not give rise to circular RNAs, indicating an additional parameter regulating their production. Overall, circularized exons are generally much larger than these skipped, uncircularized exons. To test if the length of circularized exons directly affects their circularization efficiency (CE), we utilized high-throughput and comprehensive mutagenesis of the *mrps16* exon, which revealed that exon size is a significant determinant of RNA circularization.

## Results

### Direct biochemical detection of backsplicing intermediates supports the existence of an exon-containing lariat precursor

We considered two models for the production of circular RNA in *mrps16*: that circular RNA is generated by 'direct backsplicing,' perhaps facilitated by complementary sequence (*Figure 1*, left), or through a lariat precursor (*Figure 1*, right). Each model predicts a distinct set of biochemical intermediates and byproducts: a Y-shaped intermediate in the case of direct backsplicing; and a lariat precursor and double lariat in the case of the lariat model. These structurally distinct species discriminate the models, and we reasoned that determining which of these products exists could provide initial evidence for one or both of these models. Before performing biochemical identification of these intermediates, we computationally predicted the structure of the *mrps16* pre-mRNA to determine if there were any base-pairing interactions across the circularized exon. We found that *mrps16* lacks any apparent base pairing between the regions flanking the circularized exon (*Figure 1—figure supplements 1, 2*); thus, we hypothesized that the biogenesis of this circular RNA might proceed through a lariat precursor generated by exon skipping.

As canonical lariats are of low abundance and difficult to detect in wild-type fission yeast, we used a strain of yeast (*dbr1Δ*) lacking the debranching enzyme, which catalyzes the hydrolysis of the branch point 2′–5′ linkage, the first step of lariat decay (*Ruskin and Green, 1985*). In this mutant, branched species can reach a much higher steady state concentration (*Nam et al., 1997*). We attempted to detect each of the products described above by reverse transcription-PCR (RT-PCR), Sanger sequencing all major products to determine their identities.

Using outward-facing primers, we detected the canonical lariat byproducts from linear splicing, formed by the introns of *mrps16* in *dbr1Δ* yeast (*Figure 2A*, Lanes 3–6) along with the circularized form of exon 2 in both strains (*Figure 2A*, Lanes 7–8). Although *mrps16* is not known to have exon skipping, we easily detected the exon-containing lariat resulting from exon skipping (i.e., the lariat precursor) (*Figure 2A*, Lane 10) and verified the product's sequence through Sanger sequencing (*Figure 2B*, top). Using inward-facing primers that would amplify a linear exon 1–3 splice, we did not detect an exon-skipped transcript in either strain (*Figure 2A*, Lanes 1–2). This result is in agreement with recent evidence from deep-sequencing experiments, which revealed that while exon skipping is quite pervasive in *S. pombe*, most exon-skipped transcripts are subject to nuclear decay and are thus detectable only in mutants deficient in decay (*Bitton et al., 2015*). Specifically, exon-skipped transcripts for *mrps16* can compose up to 5% of all detected *mrps16* transcripts in *S. pombe* mutants deficient in Dis3, a nuclear exosome component (*Bitton et al., 2015*; See Supp. Table S5). This represents a >30-fold increase over wild-type *S. pombe*.

To quantify the relative abundance of the *mprs16* linear, circular, and exon-containing lariat species, we performed qPCR on RNA from both wild-type and *dbr1Δ* yeast (*Figure 2—figure supplement 1*). We observed that the circular RNA is expressed at ~one-eighth the expression of the linear mRNA, and in *dbr1Δ* yeast, the exon-containing lariat is expressed at ~1/30th the expression of the linear RNA.

To test for the branched sequence indicative of the double lariat and Y-shaped intermediate (see *Figure 1*), we performed a PCR using inward-facing primers in the introns of *mrps16* (*Figure 2A*, Lanes 11–12), but we were only able to recover an unspliced product containing exon 2, which could be derived from genomic DNA, pre-mRNA, and the exon 2-containing lariat; the high abundance of these species impedes the detection of the putative double lariat/Y-shaped intermediate products. To deplete these species prior to PCR amplification, we used the restriction enzyme *Dra*I, which cuts uniquely within exon 2 of *mrps16* (see schematic in *Figure 2C*). In parallel, prior to cDNA synthesis, we treated RNA with RNase R, a 3′–5′ exonuclease, which degrades linear species but cannot degrade circular RNA or lariats, or with a mock treatment not containing RNase.

We tested the sensitivity of the exon-containing lariat and linear mRNA to *Dra*I and RNase R treatment as controls: RNase R decreases the abundance of the mRNA only, while treatment with *Dra*I decreases the abundance of both mRNA and the exon-containing lariat (*Figure 2D*, Lanes 1–8). Because of its high expression, there was still a significant amount of linear mRNA after each of these treatments, and treating with both *Dra*I and RNase R had an additive effect on its depletion (*Figure 2D*, Lanes 1–4). Using this combined depletion, we again attempted to amplify the branched sequence indicative of the backsplicing event with inward-facing primers in the introns of *mrps16*.

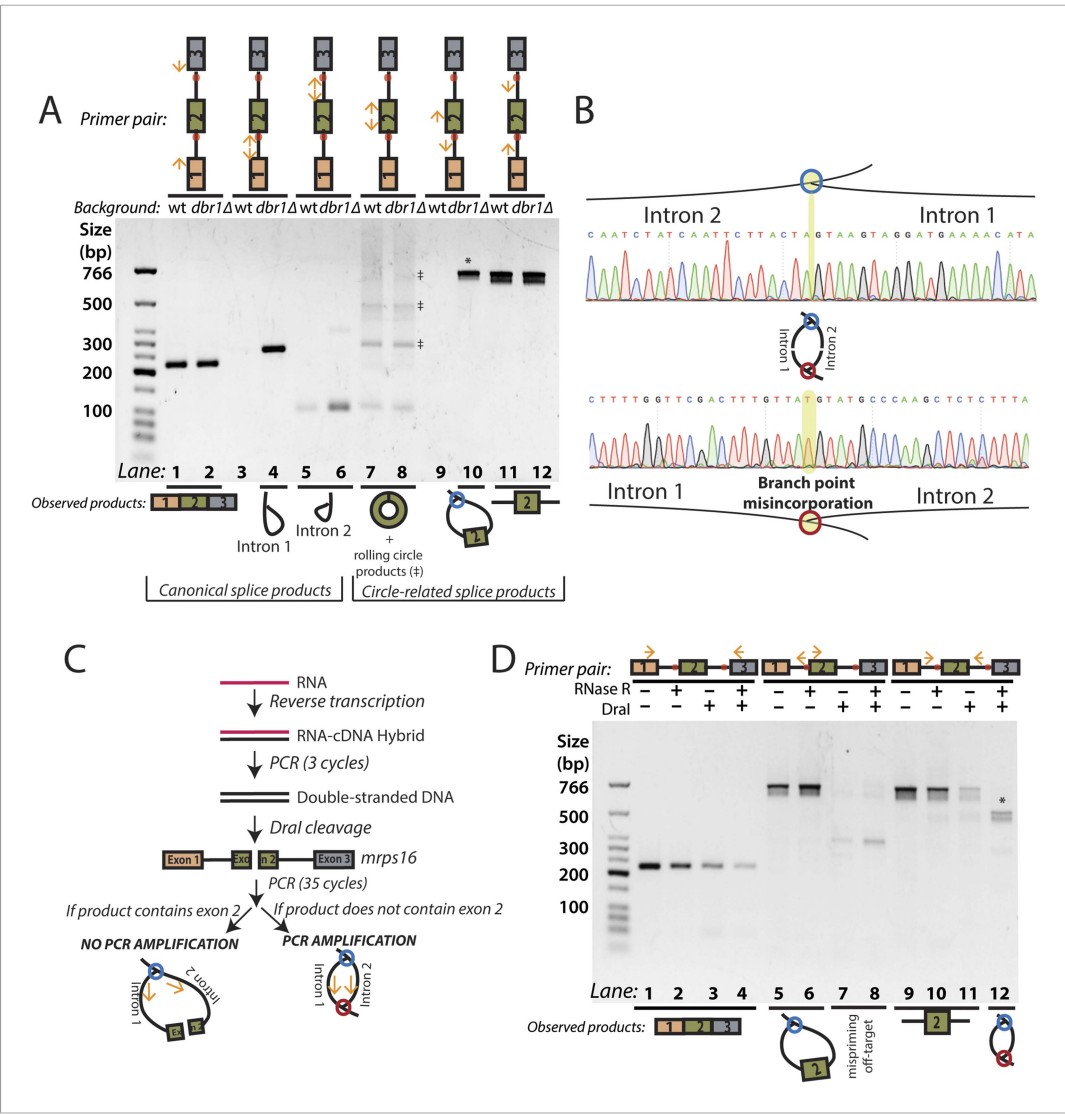

**Figure 2**. RT-PCR analysis of circular RNA biogenesis intermediates. (**A**) reverse transcription-PCR (RT-PCR) of splice products and byproducts of *mrps16 pre-mRNA* in wild-type and *dbr1Δ S. pombe*. Primers used are indicated above each lane of the gel. Red dots represent the location of branch points. The observed products are shown below the gel. Laddering of bands in lanes 7 and 8 is due to rolling-circle reverse transcription of the 181 nt *mrps16* circular RNA. (**B**) Sequencing trace for product in *Figure 2A*, lane 10 (Top) and *Figure 2D* lane 12 (Bottom) (each marked with * on respective gels). Due to the 2′–5′ linkage at the branch point, there is base misincorporation or nucleotide skipping at the branch point by the reverse transcriptase (highlighted in trace). Note, the top trace is mirrored for consistency with the orientation in the graphic representation. (**C**) Strategy to enrich for the byproduct of circular RNA backsplicing event. Since the circularized exon of *mrps16* contains a *Dra*I restriction site, we can deplete PCR products containing the exon. (**D**) RT-PCR results after RNase R and/or *Dra*I treatment (treatment indicated above the gel).

The following figure supplements are available for figure 2:

**Figure supplement 1**. qPCR analysis of *mrps16* splice isoforms.

**Figure supplement 2**. *Figure 2D*, Lane 12 cloning and sequencing results.

**Figure supplement 3**. Alignment of backsplice byproduct PCR products.

**Figure supplement 4**. Additional *Dra*I restriction digest PCR for Y-shaped intermediate product.

As before, we amplified a dominant product containing exon 2 after treating with DraI or RNase R separately. However, after the combined treatment, we discovered a new product at the expected size (518 bp) for the branched byproduct diagnostic of backsplicing (*Figure 2D*, Lane 12); this product is also faintly visible in singly-treated lanes (*Figure 2D*, Lanes 10 and 11). Sanger sequencing revealed the expected sequence for the product with a misincorporation event at the branch point, which can arise during reverse transcription of lariats (*Figure 2B*, bottom) (*Vogel et al., 1997*). We noted the 'doublet' appearance of this product, and through thorough characterization by cloning and sequencing, we determined that the two bands do not represent distinct splice isoforms and are likely a technical artifact due to anomalous migration on our gel (*Figure 2—figure supplements 2, 3*).

The enrichment of this product after RNase R treatment is strong evidence that this PCR product was derived from the double lariat rather than the Y-shaped intermediate, as the latter would be degraded by RNase R. In addition, by moving the forward primer upstream by only ~100 bp into exon 1, we were unable to detect a PCR product indicative of the Y-shaped intermediate (predicted size 634 bp) after treatment with DraI, RNase R, or both (*Figure 2—figure supplement 4*, Lanes 13–16).

## Quantitative analysis of splice isoforms demonstrates that the dominant pathway for circular RNA biogenesis in *mrps16* is through an exon-containing lariat precursor

With evidence for the existence of the putative lariat precursor and double lariat, we performed genetic tests to determine if this model is the dominant pathway for circular RNA biogenesis in *mrps16*. We cloned the *mrps16* gene in its entirety, along with 1 kb upstream and downstream into a plasmid, which allows the gene to be expressed using its natural promoter and terminator. Because *mrps16* is an essential gene, we expressed the plasmid copy of the gene in addition to its genomic copy. In *dbr1Δ S. pombe*, the plasmid expression of the gene is roughly 10 fold higher than the expression from genomic copy, likely due to the high-copy number of the plasmid. The ratio of the dominant splice isoforms of *mrps16* (i.e., circular, linear, and lariat precursor) is consistent between the genomic and plasmid copy of the gene, indicating expression from the plasmid does not bias the splicing pattern in a significant way (*Figure 3A*). In addition, the circular RNA produced from the plasmid is RNase R resistant and, notably, exhibits the same level of RNase R resistance as the genomic copy (*Figure 3B*), strong evidence that the plasmid generates a circle rather than a spliced linear concatamer of exons.

Using the plasmid copy of *mrps16*, we systematically deleted each of the splice sites and branch points in the introns of the gene and transformed these mutants into *dbr1Δ S. pombe*. As predicted by both models (*Figure 1*), qPCR confirmed that the branch point upstream of the circularized exon and the 5′ ss in intron 2 are both necessary for circularization to occur (*Figure 3C*). Some of our mutagenesis experiments were confounded by activation of cryptic splice sites in *mrps16*. For example, deletion of the 3′ ss upstream of the circularized exon has almost no impact on circular RNA production (*Figure 3C*). This is due to cryptic splicing at a nearby 3′ ss located 5 nt downstream of the original splice site (determined by Sanger sequencing, data not shown).

Both the 5′ ss of intron 1 and branch point of intron 2 are uniquely necessary for circular RNA biogenesis under the lariat model, which predicts that deleting either of these sites should disrupt circularization. On the other hand, neither of these sites play necessary roles in the direct-backsplice model; if anything, their deletion might be expected to increase circularization by decreasing competition from linear splicing. As predicted by the lariat model, deletion of the downstream branch point abolishes circular RNA biogenesis from the plasmid (*Figure 3C*), and indeed this is the only mutant in which no lariat precursor is produced above background (*Figure 3D*). In contrast, deletion of the 5′ ss of intron 1 results in cryptic splicing of a larger lariat species (*Figure 3D*), containing 74 nt of the upstream exon (*Figure 3—figure supplement 1*, Lane 11; product validated by sequencing) and also results in circular RNA production (*Figure 3C*).

Using steady state, first-order kinetics (*Figure 3E*) to model circular RNA production in each of the mutants, we tested whether our qPCR data were consistent with the kinetic predictions of the lariat model. This model predicts that the fold change in rate of the second round of splicing corresponding to a given splice site deletion ($k_{2mut}/k_{2wt}$) is equal to the relative circle:lariat ratio (plotted for each mutant in *Figure 3F*). Under this model, deletions which do not perturb the kinetics of the second round of splicing will have an unchanged circle:lariat ratio. Neglecting cryptic splicing, deletion of the

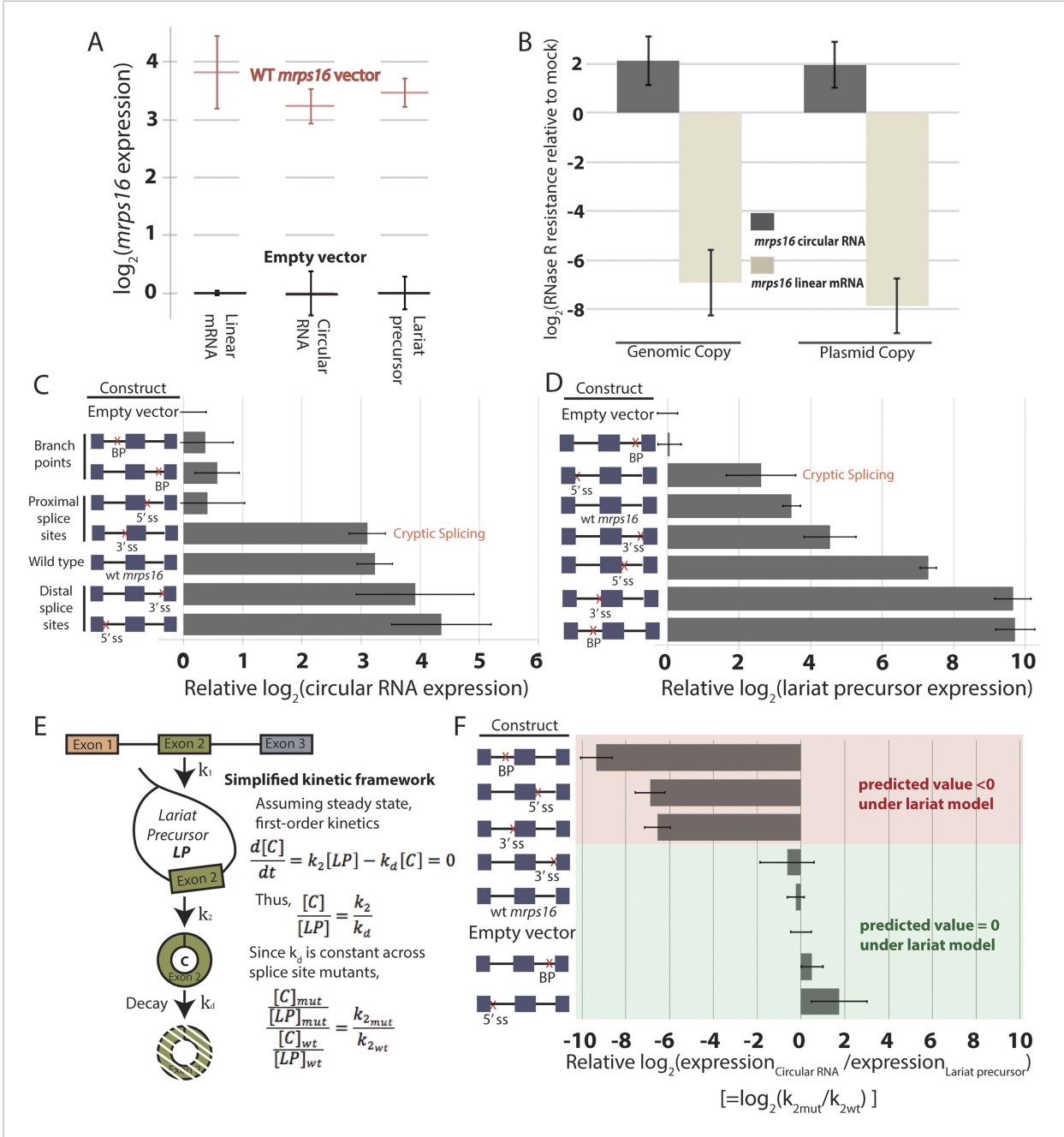

**Figure 3.** Quantitative analysis of *mrps16* splice isoforms in *dbr1Δ S. pombe* reveals circular RNA can be produced through an exon-containing lariat precursor. (**A**) qPCR quantification of splice isoforms produced by ectopically expressed *mrps16* relative to background expression of each isoform. (**B**) qPCR quantification of RNase R resistance of the endogenously and ectopically expressed *mrps16* circular and linear RNA isoforms. Positive values represent enrichment, while negative values represent depletion after RNase R treatment. (**C–D**) qPCR quantification of ectopically expressed *mrps16* circular RNA (**C**) and lariat precursor RNA (**D**) for splice site and branch point mutants relative to background. Cryptic splicing of the 3′ ss in intron 1 generates a circular RNA whose junction differs by 5 nt, and cryptic splicing of the 5′ ss in intron 1 generates a larger lariat precursor using a 5′ ss positioned 74 nt upstream of the original splice site. (**E**) Simplified kinetic framework for the production and decay of circular RNA. Under this model, the values presented in (**F**) represent $\log_2(k_{2mut}/k_{2wt})$. (**F**) Ratio of circular to lariat precursor RNA for splice site and branch point mutants relative to background ratios. Red (predicted value < 0) and green (predicted value = 0) coloring represents predictions under the lariat model. These predictions neglect any effects due to cryptic splicing. In all cases, error bars represent standard deviations from replicate experiments.

The following figure supplements are available for figure 3:

**Figure supplement 1**. Agarose gel electrophoresis of *mrps16* qPCR products.

*Figure 3. continued on next page*

*Figure 3. Continued*

**Figure supplement 2**. Effect of 5′ ss and BP double deletion on *mrps16* circularization.

**Figure supplement 3**. Analysis of *S. pombe pub1* circular RNA splicing.

5′ ss of intron 1 and the branch point and 3′ ss of intron 2 should have a constant circle:lariat ratio, and generally this association holds (*Figure 3F*, green bars). Another prediction of the lariat model is that deletions that disrupt the second round of splicing should have a dramatically lower circle:lariat ratio. As predicted, deletions of the branch point in intron 1 and flanking splice sites drastically reduce the rate of circular RNA biogenesis from the lariat precursor, which accumulates relative to the circular RNA (*Figure 3F*, red bars).

We observed that both mutants impacting the first round of splicing in the lariat model produce slightly more circular RNA than would be predicted under this simple kinetic model. Notably, the 5′ ss deletion in intron 1 fails to behave as a naive model would predict due to cryptic splicing that generates a variant exon 2-containing lariat. To determine if the lariat precursor remains the major pathway for circular RNA biogenesis in this mutant, we created an *mrps16* mutant containing both deletion of the upstream 5′ ss and downstream branch point (to halt production of the lariat precursor), and, indeed, circular RNA production is strongly abated (*Figure 3—figure supplement 2*). Still, for this double mutant and the single downstream branch point mutation (both of which are unable to produce a lariat precursor), circular RNA production is slightly higher than predicted under a model relying *exclusively* on a lariat. Thus, we cannot rule out that there may be a small population of circular RNA generated via a direct backsplice mechanism, as previously reported in vitro (*Pasman et al., 1996*; *Schindewolf et al., 1996*).

To determine whether the lariat model is unique to *mrps16*, we mutagenized another circle-forming gene, *pub1*. *pub1* contains four exons and produces two alternative circular isoforms: one composed only of exon 3 (*Wang et al., 2014*) and another, much less abundant isoform, consisting of exon 2 and exon 3 along with the intervening intron. From *dbr1Δ S. pombe*, we detected the corresponding lariat precursors to each circular RNA isoform by RT-PCR and sequenced the products to confirm (*Figure 3—figure supplement 3A*). To test whether the lariat precursor is necessary for circularization in this gene, we created a plasmid containing the *pub1* gene, with and without a deletion in the downstream branch point. Because *pub1* is non-essential, we performed these experiments in a *pub1Δ* strain to eliminate linear and circular RNA production from the genomic locus. RT-PCR and qPCR analysis revealed a large decrease in circular RNA expression after branch point deletion (*Figure 3—figure supplement 3B,C*), but circular RNA and fully spliced mRNA could still be detected, as a weak cryptic branch point was activated in the intron as evidenced by expression of the fully spliced linear isoform (*Figure 3—figure supplement 3B*, Lane 4).

Together, the existence of exon-containing lariats and their necessity for efficient circle production suggest that the dominant pathway for circular RNA biogenesis in *S. pombe* is through a lariat precursor.

## Exon skipping is not sufficient to generate circular RNA

Our findings raise the question: is production of an exon-containing lariat sufficient to produce circular RNA in *S. pombe*? There are only 14 reported examples of *validated*, stable exon-skipping events in wild-type *S. pombe* (*Awan et al., 2013*), which would be predicted to produce such lariats. Stable exon skipping could be sufficient for circular RNA production, but it is not required; as is the case for *mrps16*, the exon-skipped transcripts may be quite unstable yet produce exon-containing lariats competent for circle formation. We have chosen to test genes with stable exon-skipped transcripts because they are known to produce exon-containing lariats.

To assess the sufficiency of exon-containing lariats for circular RNA production, we used RT-PCR to test for these lariats and their putative corresponding circular RNAs (except in two cases where the exons were under 40 nt long). With both pairs of outward-facing primers, we observed bands indicative of exon-containing lariats based on their predicted migration and, in some cases, we observed larger PCR products consistent with rolling-circle reverse transcription of these lariats (examples shown in *Figure 4*); however, no circular RNA from any of these loci in either wild-type or

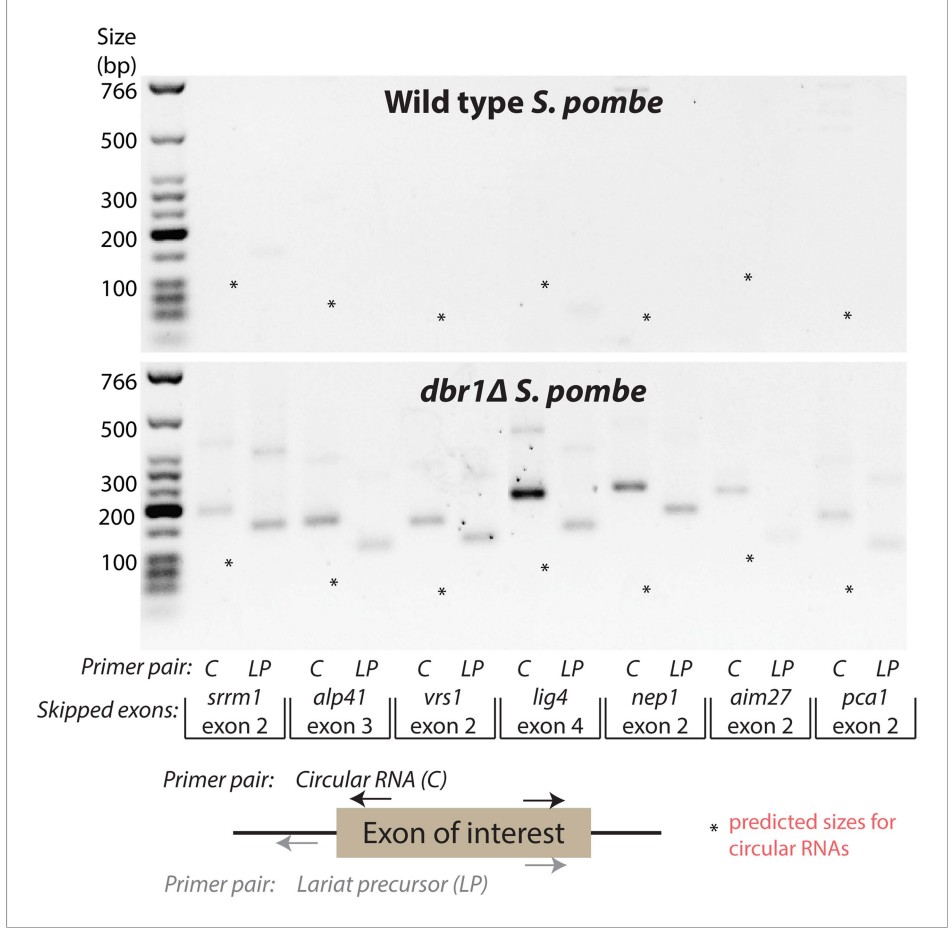

**Figure 4**. Production of exon-containing lariats is not sufficient to generate circular RNA. RT-PCR analysis of a number of genes with validated exon-skipping events in *S. pombe*. The primers pairs used are indicated below the gel. Primer pair 'C' should amplify the circular RNA or the lariat precursor if the circle is absent, whereas primer pair 'LP' should specifically amplify the lariat precursor. For each gene, asterisks indicate the expected product size for the circular RNA. In each case, only exon-containing lariats from the exon-skipping events could be detected.

*dbr1Δ S. pombe* was detected (**Figure 4**, asterisks), demonstrating that exon skipping is not sufficient for detectable levels of circular RNA.

## Unbiased, comprehensive exonic deletions reveal exon length strongly predicts circularization

We observed that skipped exons that did not produce circular RNA (**Figure 4**) were significantly shorter compared to the sizes of the predicted circularized exons in *S. pombe* (**Figure 5A**). Previous genome-wide reports in human cells have demonstrated that large exons appear to be preferentially circularized in human cells (**Zhang et al., 2014b**). However, these associations could be due to technical biases or confounding factors that are correlated with exon length (e.g., small, skipped exons may also contain exonic splicing silencers that promote exon skipping but restrict RNA circularization).

In order to test this directly, we made a 90 nt deletion in *mrps16* and found that this deletion had a dramatic effect on circular RNA abundance (total circular RNA does not differ significantly from background, **Figure 5B**). Due to the unique sequence of the deletion, we designed primers to specifically amplify the Δ90 nt *mrps16* circular species using a primer that spans the junction of the deletion (**Figure 5—figure supplement 1**). Using this primer, we quantified the levels of the Δ90 nt circle and showed the expression of this isoform was ~eightfold below endogenous circular RNA (**Figure 5—figure supplement 2**). Given the high-copy number of the plasmid (>10 per genome), this represents a >80 fold reduction in circular RNA produced by this deletion.

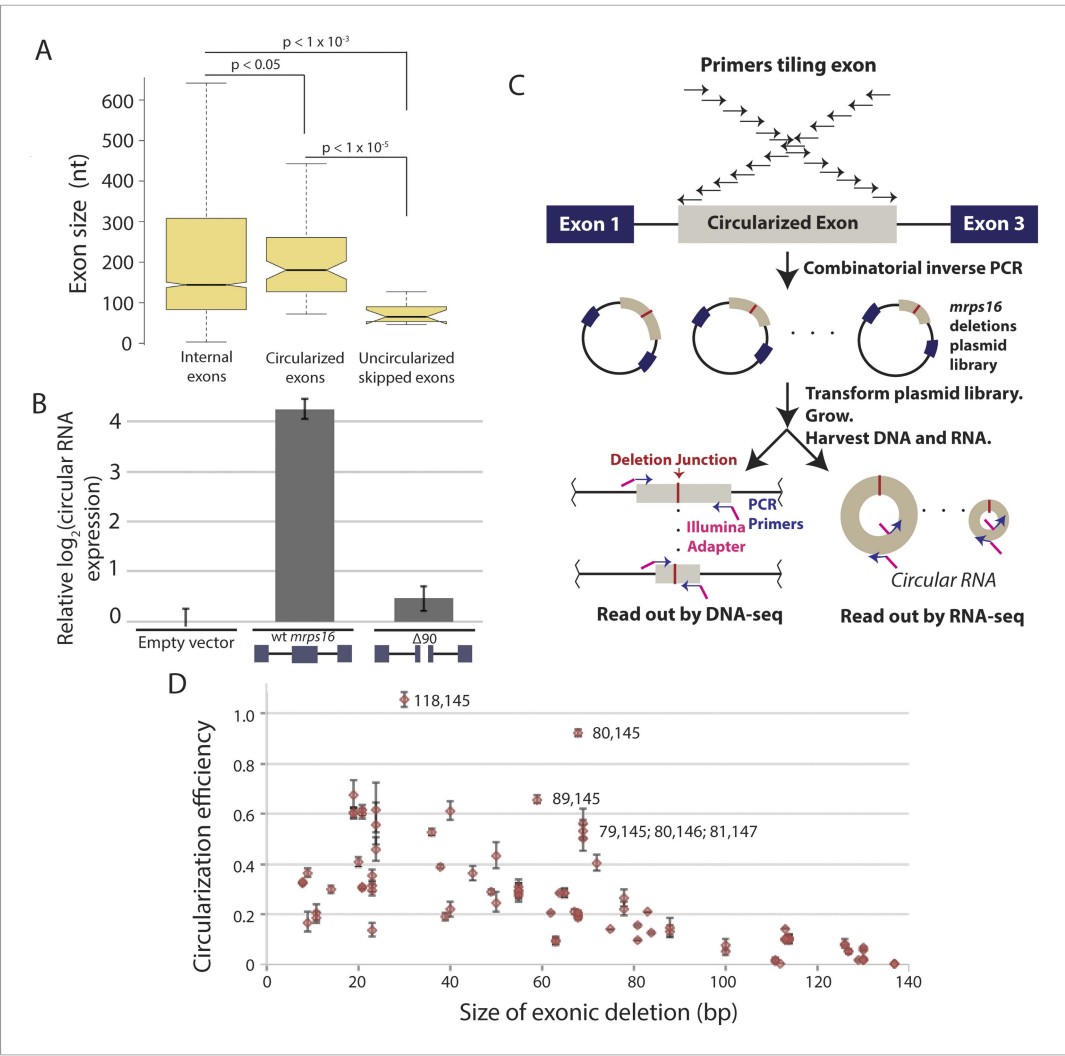

**Figure 5**. Exon size predicts circular RNA production. (**A**) Notched box plots depicting the sizes of internal exons (n = 2831), single exon circular RNAs predicted from RNA-seq (n = 86), and skipped uncircularized exons (n = 11) in *S. pombe*. The notches represent the 95% confidence interval of the median (specifically, $\pm$ 1.57 $\times$ IQR/$\sqrt{n}$, where IQR is interquartile range, and n is sample size). p-values, denoted above the plots, were calculated using a Wilcoxon rank sum test. Note only single exon circular RNAs were included in this analysis. (**B**) qPCR analysis of circular RNA production from an *mrps16* isoform lacking 90 nucleotides from the center of its second exon (*mrps16* Δ90). Total circular RNA was measured using outward-facing primers outside the bounds of the deletion and normalized to total circular RNA produced by the endogenous locus. Error bars represent standard deviations from replicate experiments. (**C**) Schematic outlining the production of *mrps16* exon plasmid deletion library and quantitative RNA-seq (qRNA-seq) method for determining isoform-specific circular RNA expression. (**D**) Circularization efficiency (CE) vs exon length as determined by qRNA-seq experiment. Outliers are noted on the plot using a parenthetical notation (x, y), where x and y represent the first and last nucleotide of deletion, respectively, with 1 representing the first nucleotide of the exon. Error bars represent two standard deviations from the mean (see 'Materials and methods' for derivation). Experiments shown in (**B**) and (**D**) were performed in wild-type *S. pombe*.

The following figure supplements are available for figure 5:

**Figure supplement 1**. RT-PCR analysis of circular RNA produced by endogenous *mrps16* and *mrps16* Δ90.

**Figure supplement 2**. Measurement of RNA extraction size bias by qPCR.

**Figure supplement 3**. qPCR analysis of splice isoforms in *mrps16* Δ90.

*Figure 5. continued on next page*

*Figure 5. Continued*

**Figure supplement 4**. qPCR analysis of splice isoform representation in exonic deletion library.

**Figure supplement 5**. Primary sequence analysis of *mrps16* exonic deletion library.

**Figure supplement 6**. *mrps16* exon 2 predicted structures.

We reasoned that this large effect could be due to a bias against retaining small RNA in our RNA extraction procedure. To address this concern, we subjected the purified RNA to a second round of extraction and purification. If this procedure is biased against small RNAs, a second round of extraction would decrease the apparent ratio of ∆90 nt (ectopic) circle to full-length (endogenous) circle. However, there was no relative depletion of small species after the second purification, supporting the conclusion that this deletion has a biological underpinning rather than being a technical artifact (*Figure 5—figure supplement 2*).

To determine whether this effect was simply caused by an increase in competitive splicing of linear mRNA in this mutant, we quantified the expression of both mRNA splice junctions but did not find a significant difference in their expression compared to *wt mrps16* (*Figure 5—figure supplement 3A*). Another potential model is that this 90 nt deletion causes a decrease in the production of the lariat precursor, leading to a decrease in observed circular RNA. In order to test this, we quantified both the abundance of the circular RNA and the lariat precursor in *dbr1∆ S. pombe*. Contrary to this model, we observe an increase in the abundance of the lariat precursor relative to the circular RNA, consistent with this mutation causing a decrease in the rate of the second round of splicing (*Figure 5—figure supplement 3B*).

Two non-mutually exclusive models could explain the decreased abundance of circular RNA from the 90 nt deletion: (1) small circles in *mrps16* do not efficiently circularize due to their size (e.g., physical constraints); (2) our deletion has removed a binding site for a *trans*-acting factor responsible for circularization. To achieve a rapid and high-throughput test of these models, we used a quantitative RNA-seq ('qRNA-seq') protocol (*Figure 5C*). We generated a pooled library of varying length deletions within exon 2 of *mrps16* by combinatorial inverse PCR and blunt ligation, using the *wt mrps16* plasmid as a template, and transformed this pool of mutants into wild-type *S. pombe*.

Next, we extracted RNA and plasmid DNA from the population. The RNA was enriched for circular isoforms by DNase and RNase R treatment and then reverse transcribed. The cDNA and plasmid DNA were PCR amplified using the same pair of inward-facing primers and cycle numbers to control for any potential sequence-specific PCR biases; barcoded-sequencing adapters were further added by PCR. Separate sequencing libraries were created at different cycle numbers to allow analysis of biases introduced by PCR.

To ensure that circular RNA species generated the vast majority of RNA-seq reads, we performed qPCR and demonstrated that: (a) the linear RNA was represented ~40 fold below the expression of the circular RNA due to RNase R treatment and (b) the lariat precursor was represented at ~300,000 fold below circular RNA, as expected since this experiment was performed using wild-type yeast (*Figure 5—figure supplement 4A*). We also showed that inward- and outward-facing primer pairs gave very similar qPCR values when used on RNase R-treated RNA, indicating both quantitate mainly circular isoform (*Figure 5—figure supplement 4B*). As an additional control, we created RNA-seq libraries with outward-facing primers, which are unable to amplify linear species.

In our library, each plasmid with a deletion contains a unique sequence within exon 2 corresponding to the deletion junction (*Figure 5C*), allowing the abundance of each circular RNA isoform to be quantified (from the RNA-seq libraries) and normalized by the representation of each plasmid in the population (from the DNA-seq libraries). By quantifying the wild-type levels of circularization, we represented our quantitative results as relative circularization efficiency (CE) for each mutant (where the value 1 represents wild-type levels of circularization from the plasmid). We tested for an effect of deletion length on CE and found a strong systematic relationship between exon length and circularization (*Figure 5D*). This trend was maintained across sequencing libraries, which vary in cycles of library amplification and primer pairs (data not shown). Note, this plot does not include the 90 nt deletion discussed above, which we analyzed separately by clonal analysis.

While exon length is significant in our model, it is not completely predictive of CE, indicating the existence of other factors that could potentially affect circularization. This library can contain multiple representatives for any given deletion size and for any given deletion of a primary sequence motif. This allows us to rapidly test first order predictions that efficient circle production relies on the presence of a trans-acting factor binding site. However, our data revealed no relationship between the location of a deletion (i.e., its inclusion of any primary sequence motif), and its representation in the library. Thus, we were unable identify a primary sequence required for circularization (see *Figure 5—figure supplement 5*).

Analysis of outliers in our *mrps16* mutagenesis data revealed that essentially all data points with higher-than-expected rates of circularization represented deletions in a region at the 3′ end of the exon that is predicted to be rich in secondary structure. A hairpin, spanning from 135 to 159, is predicted to form with high probability, and deletions that remove half of this hairpin are responsible for the most significant outliers (*Figure 5—figure supplement 6*). These outliers suggest a model in which secondary structures may limit the flexibility of the exon: removing pieces of these secondary structures can destabilize them, increasing the effective length of the circularized exon.

We predicted the minimum free energy (MFE) structures of exon 2 for several of these deletions and found that these deletions maintained the apparent, end-to-end length of the exon and in some cases actually increased the apparent length (*Figure 5—figure supplement 6*). We also found that deleting the entire hairpin (89,171) negated the enhancement in circularization (not shown), and in agreement, the predicted structure for this deletion contains far fewer free nucleotides in this case (17 vs 37 for wt) (*Figure 5—figure supplement 6*). This indicates that outliers marked in *Figure 5D* do not simply remove a repressor's-binding site.

Despite this suggestive data that exonic structure is important for circular RNA production in *mrps16*, we were unable to predict CE by computational structure prediction alone (not shown). This may reflect an inability of structure prediction methods to accurately predict in vivo RNA secondary structure or, instead, may reflect additional yet-to-be discovered regulatory features.

## Discussion

Circular RNA is a pervasive aspect of eukaryotic gene expression, and recent work has begun to shed light on the regulation and biogenesis of these still largely mysterious RNAs (*Ashwal-Fluss et al., 2014*; *Liang and Wilusz, 2014*; *Zhang et al., 2014b*; *Conn et al., 2015*). Current models for the production of circular RNAs posit that RNA secondary structures formed by inverted sequences in flanking introns are necessary elements for circularization, but this model lacks completeness, as circular RNAs can be found in genes that lack extensive complementary sequences. This is abundantly clear in lower eukaryotes whose genomes are almost devoid of repetitive sequences (*Richard et al., 2008*).

Using *S. pombe* as a model system, we provide thorough mechanistic evidence that circular RNA can be generated through an exon-containing lariat precursor, a longstanding, yet unsubstantiated model in the field (*Zaphiropoulos, 1996*; *Surono et al., 1999*). Based on splice site mutagenesis experiments, we find that lariat production is important for detectable production of *mrps16* circular RNA. However, a lariat precursor may not be strictly necessary, as suggested by our data and in vitro work in mammalian and yeast splicing extracts that have shown it is possible to directly backsplice a circular RNA in the absence of competing splice sites (*Pasman et al., 1996*; *Schindewolf et al., 1996*). Rather, it is more likely that the lariat production contributes to backsplicing catalysis by positioning splice sites. In addition to this catalytic function, lariat production may also serve to prevent competing splice sites from operating. In other words, within the context of an exon-containing lariat, the only available splice sites are those bordering the circularized exon, and the upstream branch point, now 'orphaned' by lariat formation, must attack the downstream splice donor. In this way, the exon-containing lariat may provide a microenvironment for the splicing of circular RNA.

A hypothesis generated by a model in which circular RNA is spliced from an exon-containing lariat precursor is that inhibiting lariat degradation should lead to increases in circular RNA expression. Recently, Bähler and colleagues published a broad RNA-seq study of several whole transcriptome profiles of wild-type and *dbr1Δ S. pombe*, and the authors note a ~3–4 fold global increase in circular RNA in *dbr1Δ S. pombe* (*Bitton et al., 2015*). However, upon analyzing *mrps16* specifically, we observed significant variation in the *mrps16* circular:total splice isoform ratios within the wild-type RNA-Seq biological replicates due to the very low number of reads (*Bitton et al., 2015*, Supp. Table S11).

Thus, while there appears to be a modest increase in the *global* levels of circular RNA in the debranching mutant, the effects on particular genes fail to reach statistical significance.

We have also found that production of an exon-containing lariat is not sufficient for circularization, just as inverted repeats are not sufficient for circular RNA production in mammals (*Zhang et al., 2014b*), suggesting other regulatory mechanisms responsible for its production. Supporting the idea that factors beyond the presence of an exon-containing lariat are required for circular RNA production, we have found that, in general, uncircularized skipped exons are much smaller than the circular exons, hinting that exon size may have a direct effect on the efficiency with which an exon circularizes. Using *mrps16* as a model gene, we tested this directly with a library of exonic deletions and found a systematic decrease in circularization with a decrease in exon size. Although there is a strong effect of exon size, circularization is presumably influenced by additional factors such as topological effects due to intronic sequence and combinatorial effects of RNA-binding proteins. Indeed, we have found that intronic deletions can have a pronounced effect on RNA circularization. None of these deletions (which range from 133 to 329 bp long) affect circularization as dramatically as deleting the branch point in either intron (SB + JS, unpublished data).

We have previously identified circular RNAs in genes in *Saccharomyces cerevisiae* (*Wang et al., 2014*) that are known to generate exon-skipped linear isoforms (*Hossain et al., 2011*; *Egecioglu et al., 2012*), lending support to the possibility that this mechanism extends beyond *S. pombe* to other eukaryotes. As an example, the three exon gene *SUS1* has an exon-skipped linear transcript that is rapidly decayed (*Egecioglu et al., 2012*). Substitution of the *SUS1* gene with a linear cDNA copy results in temperature sensitivity and histone H2B deubiquitination defects (*Hossain et al., 2011*). This was suggested to be due to a functional role of an alternatively spliced linear isoform, but could be due to a circularized isoform derived from the seemingly non-functional and rapidly-decayed exon-skipping event. Indeed, it may be a general theme that short-lived exon-skipping events may give rise to long-lived circularized exons, which may play a productive role in cells.

As intron size increases, it may be that exon-containing lariats no longer have a strong topological effect. This may limit the utility of this mechanism in higher eukaryotes, which may have adopted inverted sequences in an effort to produce local topological changes. However, as alluded to earlier, lariat production may serve an additional function by isolating exons for circularization, implying that these mechanisms may work in tandem. The existence of the lariat precursor model in lower eukaryotes suggests a possible mechanism in which circular RNA arose as a byproduct of exon skipping and that, as intronic architecture changed, so did mechanisms for maintaining expression of these molecules.

## Materials and methods

### Strains and media

The following *S. pombe* strains were obtained from Bioneer Inc. (Alameda, CA): 'wild-type' strain ED668 (*h+ ade6-M216 ura4-D18 leu1–32*) and its deletion derivatives *dbr1Δ* and *pub1Δ*. Strains were grown in yeast extract with supplements (YES) media when grown without selection. Selection was performed in Edinburgh Minimal Media (EMM) in the absence of leucine (EMM-Leu).

### *mrps16* and *pub1* expression vectors and mutagenesis

The base vector pW1336 was created from pREP41/GFP-Osh2 (gift of Masayuki Onishi) by removal of the nmt1 promoter and insert gene by *SphI-SacI* digest and replacement with an oligonucleotide polylinker. It contains ARS1 from *S. pombe*, LEU2 from *S. cerevisiae*, and ampicillin-resistance. *S. pombe* genomic DNA was amplified using the primers TTTCGGCGCGCCCTCTTTTTTTAAGA AAATTTCGTTGCTTG and GACGTTAATTAAATCCAGGAATGTTTTTGATGAGCACTAG (for *pub1*), and TCTAGGCGCGCCTAATTCCGTATTGATGGAG and AGGATTAATTAAGCTCTCTTGGTTGTGCAATCAG (for *mrps16*), and cloned as *AscI-PacI* fragments into pW1336. The inserts were validated by Sanger sequencing.

Plasmid deletions were generated using QuikChange Lightning Mutagenesis Kit (Agilent Technologies, Santa Clara, CA) according to the manufacturer's instructions (primers shown in *Supplementary file 1*). Mutations were validated by Sanger sequencing.

For generating the exonic deletion library, inverse PCR was performed using ~1 ng of plasmid template per 50 µl PCR reaction using Phusion polymerase (New England Biolabs [NEB], Ipswich, MA) with high-fidelity buffer according to the manufacturer's instructions (primers shown in

*Supplementary file 1*). Prior to PCR, the primers were phosphorylated using T4 polynucleotide kinase (PNK) (NEB) at 10 µM primer concentration in 1× T4 DNA ligase buffer. After PCR, these products were pooled, digested with *Dpn*1 for 4 hr at 37°C to remove vector template, and concentrated by isopropanol precipitation. The concentrated products were then run on a 1% agarose gel for size selection. The 12 kb was visualized using blue light and extracted using a clean razor blade. The product was purified using Qiagen Gel Extraction kit according to manufacturer's instructions. 50 ng of the purified products was self-ligated using 400 U T4 DNA ligase (NEB) to circularize the vectors in 20 µl at 16°C overnight. 4 µl of the reaction products were transformed into *Escherichia coli* TOP10 chemically competent cells (Life Technologies, Carlsbad, CA) according to manufacturer's instructions, and transformed cells were plated on LB + carbenicillin plates. After 17 hr of growth, 5 ml of LB was added to plates and colonies were pooled by scraping (~1500 colonies obtained). The bacteria were pelleted and plasmids were isolated using the Qiaprep Spin Miniprep kit according to the manufacturer's instructions (Qiagen, Venlo, Netherlands).

## RNA isolation
Total RNA was isolated from log-phase yeast using an acid phenol-chloroform extraction. After extraction and ethanol precipitation, RNA was treated with TURBO DNase (Life Technologies) and purified and concentrated using the Purelink RNA-mini kit (Ambion, Austin, TX).

## RNase R treatment
1 µg of total RNA was treated with 5 U RNase R (Epicentre, Madison, WI) in 10 µl total reaction volume at 37°C for 30 min. The RNA was then reverse-transcribed without purification with the addition of 4 µl 5× supplement buffer (250 mM Tris pH 8, 125 mM KCl, 15 mM MgCl$_2$) and 2 µl 0.1 M DTT to provide necessary conditions for the RT reaction.

## RT-PCR
In all cases, 1 µg of total RNA was reverse transcribed with random hexamers using 100 U Maxima H-minus reverse transcriptase (ThermoFisher Scientific, Waltham, MA) according to the manufacturer's instructions.

PCRs were performed using Platinum Taq hot-start polymerase (Life Technologies). For all PCR reactions, 1.5 µl of the unpurified RT-reaction was used per 50 µl reaction volume. All RT-PCR reactions were performed using the recommended cycling protocol for 35 cycles.

RT-*Dra*I-PCRs were assembled as normal RT-PCRs, but an initial PCR phase (3 cycles) was used to produce double-stranded cDNA. The samples were then removed from the thermocycler and placed on ice. 20 U of *Dra*I (NEB) was added directly to the 50 µl PCR reaction, and the reaction was allowed to incubate at 37°C for 1 hr. From there, the PCR was resumed for 35 additional cycles without purification.

qPCR reactions were assembled as 10 µl reactions using AccuPower 2X GreenStar qPCR Master Mix (Bioneer) and Power SYBR Green Master Mix (Life Technologies) with 0.3 µl of template used per reaction. We performed qPCRs on an ABI 7900HT using following cycling protocol: 50°C for 20 min, 95°C for 10 min, (95°C for 15 s and 60°C for 30–60 s) × 45 cycles, followed by a dissociation stage.

## RNA structure prediction
RNA structures and dotplots are the predicted MFE structures determined by NUPACK (*Dirks and Pierce, 2004*; *Dirks et al., 2007*; *Zadeh et al., 2011*) with default RNA settings at a temperature of 30°C.

## *dbr1*Δ and wild-type RNA-seq analysis
Adapters were trimmed from reads using CutAdapt (*Martin, 2011*). Using our custom pipeline (*Szabo et al., 2015*), we aligned the reads from (*Bitton et al., 2014*) (NCBI Gene Expression Omnibus accession number GSE50246) to the *S. pombe* transcriptome and determined the circular junctional reads derived from single exons (analysis shown in *Figure 5A*). Only junctions with a probability of greater than 0.7 were considered here.

## qRNA-seq sequencing library preparation

Wild-type *S. pombe* was transformed with the plasmid library (see 'Expression vectors and mutagenesis' for details on the creation of the plasmid library) according to standard protocol, and transformants were plated on EMM-Leu to select. ~2000 colonies were obtained and scraped using 5 ml EMM-Leu. Cells were mixed to homogenize the sample and 2 ml of cells were used for RNA isolation and 2 ml for DNA isolation. RNA was isolated by acid-phenol chloroform extraction and was treated with both RNase R (Epicentre) and Turbo DNase (Ambion) prior to cDNA synthesis. Plasmid DNA was isolated using NucleoSpin Plasmid Miniprep Kit (Macherey–Nagel, Bethlehem, PA) using a published supplementary protocol for purification of plasmid DNA from yeast. To enrich for plasmid DNA in the sample, we also treated with Exonuclease V (NEB) to degrade linear DNA fragments.

Sequencing libraries were produced using a PCR approach based on a previously reported method (*Zhang et al., 2014a*) with some key differences. In our protocol, PCRs were performed using Phusion polymerase (NEB) for 5 cycles in an initial amplification stage using tagged gene specific primers followed by n-5 cycles for the addition of the sequencing adapters and barcodes, where n = 25, 30, or 35 cycles (each library receiving a unique barcode in order to reference cycle number). Primers for the latter PCR stage were exactly as reported in Zhang et al., and the primers were obtained directly from the Li Lab (Sanford University; Stanford, CA). Libraries were pooled and spiked with PhiX to a concentration of 25%. Sequencing was performed on an Illumina MiSeq.

## qRNA-seq analysis

Reads were trimmed using CutAdapt (*Martin, 2011*) and only reads greater than 40 nucleotides after trimming were used for analysis. We aligned reads, using Bowtie 1 (*Langmead et al., 2009*), first to an index containing an unmutated *mrps16* sequence, and then the unaligned reads were aligned using a custom-built index containing all possible deletions of the *mrps16* exon that could be generated using our combinatorial, outward-facing PCR strategy. In both cases, we limited the alignments to at most 3 mismatches using the Bowtie '–v 3' setting. Since the primers were not purified after synthesis or before our combinatorial PCR, incomplete primer synthesis could generate mutants in our plasmid library, and we accounted for this in generating our index. This leads to a number of variants differing by only a few nucleotides.

After aligning to these indexes, we determined the number of reads derived from each deletion isoform in each sequencing library, filtering out isoforms with less than 200 reads derived from the plasmid-based DNA-seq library. We converted these read counts to circularization efficiency (CE) using the following equation:

$$CE = \frac{n_c/(n_{wt} \times s)}{n_v/n_{v_{tot}}},$$

where $n_c$ is the number of reads derived from a given circular RNA isoform, $n_{wt}$ is the number of reads derived from wild-type circular RNA (from background genomic expression), and s is a scaling factor that corrects for the increased average copy number of the plasmid, which we assume is not altered by the exonic deletion. This value is determined from qPCR data from cells transformed with a *wt mrps16* plasmid (s = 19 in wild-type *S. pombe*). This value is then normalized by the representation of the given vector isoform in the plasmid library ($n_v/n_{vtot}$), where $n_v$ is the number of reads derived from a given plasmid isoform, and $n_{vtot}$ is the total number of reads in the plasmid-derived library (after removing reads that align to the wild-type sequence).

Error bars in *Figure 5D* were generated using the relationship that $n_{wt}$ and $n_c$ are independent Poisson random variables and therefore, X: = $n_c|(n_{wt} + n_c)$ is bionomial with parameters probability of success B:= $n_c/(n_c+n_w)$ and total trials $n_c+n_w$. Standard binomial confidence intervals were assigned to this quantity, and the 95% CI was propagated to A: = $n_v/n_{vtot}$ (expression A) using the transformation f(A) = 1/A. Expression B can be converted to $n_c/(s * n_{wt})$ using the transformation g(B) = B/(1 − B)/s.

Note that, CE = f(A) * g(B).

The total variance of ln(CE) is obtained by combining a normal approximation for B (corresponding to $\hat{p}$ in a binomial model) and the delta method, which states that for sufficiently regular functions h, var(h(x)) = var(x) h′(x)^2.

Therefore,

$$var(\ln(CE)) = var(\ln(f(A)g(B))) = var(\ln(f(A))) + var(\ln(g(B))),$$

and applying

$$var(\ln(f(A))) = var(\ln(1/A)) = var(\ln(A)) = var(A)*[\ln(A)]^2 = var(A)*1/A^2,$$

where var(A) is estimated as $A(1-A)/n_{vtot}$.

$$var(\ln(g(B))) = var(\ln(B/(1-B)/s)) = var(B)*[\ln(B/(1-B)/s)]^2 = var(B)*1 \Big/ (B(1-B))^2,$$

where var(B) is estimated as $B(1-B)/(n_{wt} + n_c)$.

Using these relationships, we computed the sd = sqrt(var(ln(CE))).

Finally, we transformed the approximate 95% CI for ln(CE) (its point estimate plus or minus 2 sd) to the 95% CI for CE by exponentiating the upper and lower bounds of the confidence interval for ln(CE), resulting in the error bars on the plot.

## Acknowledgements

We thank Rui Zhang and Jin Billy Li (Stanford) for providing the PCR primers used for the generation of RNA-seq libraries. We also thank Linda Szabo for adapting her alignment pipeline for use with *S. pombe*.

## Additional information

### Funding

| Funder | Grant reference | Author |
|---|---|---|
| National Cancer Institute (NCI) | R00, CA168987-03 | Julia Salzman |
| Donald E. and Delia B. Baxter Foundation | | Julia Salzman |
| Stanford University | Stanford Graduate Fellowship | Steven P Barrett |
| Lucille P. Markey Charitable Trust | Lucille P. Markey Biomedical Research Fellowship | Steven P Barrett |
| Alfred P. Sloan Foundation | | Julia Salzman |

The funders had no role in study design, data collection and interpretation, or the decision to submit the work for publication.

### Author contributions

SPB, Conception and design, Acquisition of data, Analysis and interpretation of data, Drafting or revising the article; PLW, Conception and design, Drafting or revising the article, Contributed unpublished essential data or reagents; JS, Conception and design, Analysis and interpretation of data, Drafting or revising the article

## Additional files

### Supplementary file

• Supplementary file 1. List of primers used for cloning in this study.

### Major dataset

The following previously published dataset was used:

| Author(s) | Year | Dataset title | Dataset ID and/or URL | Database, license, and accessibility information |
|---|---|---|---|---|
| Bitton DA, Rallis C, Jeffares DC, Smith GC, Chen YY, Codlin S, Marguerat S, Bähler J | 2014 | LaSSO, a strategy for genome-wide mapping of intronic lariats and branch-points using RNA-seq | http://www.ncbi.nlm.nih.gov/geo/query/acc.cgi?acc=GSE50246 | Publicly available at NCBI Gene Expression Omnibus (Accession No: GSE50246). |

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
