## [Decision Letter]

Thank you for sending your work entitled “Circular RNA biogenesis can proceed through a lariat intermediate and is dictated by exon size” for consideration at *eLife*. Your article has been overall favorably evaluated by James Manley (Senior editor) and three reviewers, one of whom, Douglas Black, is a member of our Board of Reviewing Editors. Based on these reviews, summarized below, we will be happy to consider a revised submission addressing the points raised.

Summary:

This study by Salzman and colleagues examines the biogenesis of circular exon RNAs. This area is very topical. Although back-splicing to generate exon circles has long been described, recent studies have shown that these circles are common and often abundant gene products of largely unknown function. The authors here examine structural features that give rise to a back-spliced exon circle in *S. pombe* derived from exon 2 of the *mrps16* transcript. Previous results with metazoan exons indicated that RNA base-pairing interactions between sequences flanking an exon can enhance circle formation. However, many exons including *mrps16* exon 2 do not appear to have such an intron pairing interaction. Instead, splicing of exon 1 to exon 3 leads to a lariat intron containing exon 2 and the authors show that this serves as the precursor for circle formation. This pathway predicts that back-splicing of exon 2 within the lariat will yield a doubly branched RNA product containing the two flanking introns. Using a series of RT/PCR reactions with primers diagnostic of different RNA structures, as well as assaying the sensitivity of the products to various nucleases, they define many of the expected RNA products of the *mrps16* gene. RT/PCR around the lariat loop in a debranching deficient strain indicates that the exon-skipped lariat is produced. The Exon 1-Exon 3 spliced product is not detected, presumably due to nuclear or cytoplasmic decay (although it would be nice to show this). A final pair of primers produces a product with both branchpoint-5' splice site junctions, indicating that the doubly branched RNA lacking exon 2 is indeed produced. Mutation of the branchpoint upstream from exon 3 greatly reduces formation of the exon 2 circle indicating that formation of a lariat this site is a prerequisite of circle formation. Mutation of some of the other sites is complicated by cryptic splice site activation. Experiments with another circle derived from the *pub1* gene indicate that it is also formed from a lariat intermediate. Analysis of a debranchase deficient strain indicates increased circle formation genomewide in the absence of the debranching enzyme, including from the *mrps16* locus. However, the authors show for several other cases in *S. pombe* that exon skipping does not always lead to circle formation. They find that the skipped exons that do not produce circles are shorter than circle generating exons. A 90 nucleotide deletion within exon 2 greatly reduced circle formation. To further test the effect of exon length, they make a library of deletions within exon 2 and measure circularization relative to deletion size, finding a very rough increase in circle formation with longer exons. From these data, they conclude that both the formation of an exon-containing lariat by exon skipping, and the exon length contribute to frequency of back-splicing to form an exon circle.

All the reviewers found these data very interesting and that the authors were judicious in their interpretation of them. The identification of a doubly branched product from exon circularization is a striking and new finding that will influence the interpretation of other studies of exon circles. These data serve as a nice counterpoint to a recent paper in Cell, reporting that base pairing interactions between flanking introns can contribute to exon circle formation. As the authors point out, the exon-containing lariat may be a relevant feature when the introns are short, and secondary structure may contribute in some cases where the introns are long. However, it is not ruled out that the base-pairing in introns leads to exon skipping and hence to a similar pathway to that described here. Neither a base pairing interaction nor an exon-containing lariat seem to be a universal requirement; There still seem to be many circles that don't have either one. The exon length data was much less convincing, with few solid conclusions. The demonstration of the exon containing lariat as the precursor to the exon circle and the identification of the double branch product after circularization are results of significant interest for RNA biology. There are a number of aspects of this study that the reviewers felt should be improved.

Essential revisions:

One concern is the use of quantitative terminology that is inappropriate in the absence of standard curves. For example, in the subsection headed “Direct biochemical detection of backsplicing intermediates supports the existence of a lariat intermediate”, the statement “remarkably high levels” appears to be solely based on a robust band observed after 35 cycles of PCR. As all other products on this gel are derived from different primer combinations, we cannot tell whether the level is “remarkably high” as we have no relevant comparison. While quantitative assessment may not be essential to support all of the conclusions, efforts should be made to report on the actual amounts of some of the products they describe. It would strengthen our understanding of these processes to know the relative amount of RNA produced by exon-skipping and circularization compared to the exon-inclusion pathway. Where measurement is not possible, quantitative language should be avoided.

Given the nature of the PCR assay it is important that the identity of each product is confirmed by sequencing. This has been done in some cases, but are all the band identities validated in this way? The bands in Figure 5 are dismissed due to incorrect size, but what are these products? In several figures, bands are doublets (e.g. Figure 2, lane 12), but there is no mention in the text of whether two different products were identified by sequencing. This could have implications for the mechanism and should be addressed at least for the double lariat.

The correlation of circle formation with deletion size within exon 2 seems very rough at best (Figure 6). Moreover, there are other possible effects of these mutations that do not seem to have been explored. First, it is not clear that circularization should vary continuously with length. Perhaps there is a minimum size below which an exon cannot be circularized. In fact looking at the plot, there might be such a discontinuity as their deletion reaches 100 bp. The authors also do not control for other effects that these deletions might have. One could imagine that a deletion simply removes a specific sequence required by the splicing machinery to successfully back-splice. This possibility might be controlled for by adding back different sequences to restore the length while maintaining the loss of elements specific to exon 2. Conversely, a deletion might eliminate exon skipping altogether. If the mutant exon 2 is very efficiently spliced, there may be no substrate for back splicing. This also could be examined.

Although the authors do a careful and thorough job of laying out their arguments, the paper is not an easy read. A number of choices of terminology and presentation are not standard usage and will confuse readers. These must be fixed both to adhere to the actual biochemical relationships of the molecules, and to avoid confusion with other previously established uses for the same terms. Major examples are listed here with others in the minor comments.

Figures:

In Figure 2, the brackets showing the rolling circle RT-RCR products in lane 7 and 8 make lane 9 unreadable. Even though there is no signal in that lane, the reader should be able to see lane 9. Asterisks next to the bands in lanes 7 and 8 should work fine.

Figure 2 has a mirror image presentation of one sequence. This is confusing to assess and violates conventions against using mirror image letters and reading sequence 5' to 3'. Same for Figure 3—figure supplement 3. One might justify not following these conventions if it clarified some aspect of the data, but in this case it would be clearer to follow them.

In Figure 5, it seems like the red dashed lines are covering the place in the gel where the reader is supposed to assess the absence of signal. This is not possible with the dashed lines. An asterisk next to the lane would suffice to mark the position while still allowing the reader to see that there is no band there.

Terminology:

In the splicing field it is standard usage that the lariat intron and the spliced exons are both products of splicing. Because there are two catalytic steps, there are also intermediates called the free 5' exon and the “lariat intermediate”, which is the lariat intron still joined to exon 2. The use of the term “lariat intermediate” throughout this paper is confusing on two counts. First, the term has a different meaning from standard usage in the field. Second, the authors work quite hard to establish this exon-skipped lariat product as the precursor molecule for exon circularization. Possible alternative language for heading of the second section in the Results: “Quantitative analysis of splice isoforms demonstrates that the dominant pathway for circular RNA biogenesis in *mrps16* uses the lariat product of exon skipping as a precursor in a second round of splicing”.

Finally what is a “terminated fork”? The direct backsplicing mechanism would involve a Y-shaped intermediate, similar to that produced by trans-splicing, instead of a lariat intermediate, and in the second step would produce a circle and a Y-shaped product instead of a lariat product.

Minor points:

1) In Figure 6–figure supplement 4, the authors show a secondary structure predicted for the exon. It would be interesting to know more about this. Do finer grained mutations that disrupt it affect circularization? As the authors state, a secondary structure would shorten the effective distance between the exon ends, but it may also possibly interfere with assembly of spliceosomal components or alter their bound conformation, which perhaps could positively affect the ability to back-splice. It would be helpful to resolve more clearly if the structure affected circle formation or not.

2) Is the ∆90 mutation plotted on Figure 6D?

3) It may be the exposure of the gel, but in Figure 3—figure supplement 3 the bands in B don't seem to agree with the bar graph in C. It appears that there is more of the BP-mutated one-exon circle relative to wild type than of the BP-mutated two-exon circle relative to wild type. The bar graph shows the opposite.

4) In Figure 3—figure supplement 3, what do the two shades of gray mean in the bar graph?

---

## [Author Response]

*All the reviewers found these data very interesting and that the authors were judicious in their interpretation of them. The identification of a doubly branched product from exon circularization is a striking and new finding that will influence the interpretation of other studies of exon circles. These data serve as a nice counterpoint to a recent paper in Cell, reporting that base pairing interactions between flanking introns can contribute to exon circle formation. As the authors point out, the exon-containing lariat may be a relevant feature when the introns are short, and secondary structure may contribute in some cases where the introns are long. However, it is not ruled out that the base-pairing in introns leads to exon skipping and hence to a similar pathway to that described here. Neither a base pairing interaction nor an exon-containing lariat seem to be a universal requirement; There still seem to be many circles that don't have either one. The exon length data was much less convincing, with few solid conclusions. The demonstration of the exon containing lariat as the precursor to the exon circle and the identification of the double branch product after circularization are results of significant interest for RNA biology. There are a number of aspects of this study that the reviewers felt should be improved*.

We appreciate the expert and insightful interpretation of our paper. We have attempted to clarify a very interesting point raised above: because of both nuclear and cytoplasmic surveillance mechanisms, exon-skipped products, from yeast to humans, can be highly unstable. Hence, the presence of an exon-skipped linear RNA is not a prerequisite for the existence of exon-containing lariat. We have revised the draft to include a discussion of this, because as you point out, it may be important for broader inference on mechanisms responsible for circular RNA biogenesis. Finally, as detailed below, we have clarified our interpretation of our mutation analyses leading us to find an effect of exon length on circular RNA abundance.

*Essential revisions*:

*One concern is the use of quantitative terminology that is inappropriate in the absence of standard curves. For example, in the subsection headed “Direct biochemical detection of backsplicing intermediates supports the existence of a lariat intermediate”, the statement* “*remarkably high levels*” *appears to be solely based on a robust band observed after 35 cycles of PCR. As all other products on this gel are derived from different primer combinations, we cannot tell whether the level is* “*remarkably high*” *as we have no relevant comparison. While quantitative assessment may not be essential to support all of the conclusions, efforts should be made to report on the actual amounts of some of the products they describe. It would strengthen our understanding of these processes to know the relative amount of RNA produced by exon-skipping and circularization compared to the exon-inclusion pathway. Where measurement is not possible, quantitative language should be avoided*.

Thank you for pointing this out. We have revised our language and added quantitation when possible, which includes qPCR quantification of the relative abundance of various *mrps16* splice isoforms (linear mRNA, circular RNA, and lariat precursor) in wild type and *dbr1∆* yeast (Figure 2—figure supplement 1).

Your request and a recent study by Bahler and colleagues (3) that was published while our work was in review motivated us to examine the concordance between our RNA-Seq analysis and our qPCR studies of WT and debranching mutants. [3] includes a broad RNA-Seq study of several whole transcriptome RNA-seq profiles of WT *S. pombe,* as well as several splicing mutants. Like our analysis of the Bahler groups’ 2014 data which we discussed in our manuscript, [3] find a global increase in expression of circular RNA in the same dataset of lariat debranching on circular RNAs (4).

We sought to confirm this result and our qPCR shows a slight (∼50%), but statistically non-significant increase in abundance of circular RNA compared to the linear isoform in the debranching mutant. In addition, by examining the recent analysis in [3], we observed significant variation in the *mrps16* circle: total splice isoform ratio within the WT RNA-Seq biological replicates. For example, in one case, the *mrps16* circle:total splice ratio is actually higher in WT than in either of the debranching mutant replicates which varies from 0.13 to 0.18 (see [3], Supplementary Table S11, line 8).

These findings are in contrast to our initial RNA-seq analysis which found a statistically significant 6-fold difference in this ratio between the two strains. We have repeated our qPCR to assess whether input RNA mass RNA into the RT step, different RT priming strategies, different reverse transcriptases, or different qPCR mixes are responsible for this effect. In none of these cases do we see a 6-fold effect.

Our RNA-seq analysis also found a *global* 6-fold increase in the circle:linear ratio in *dbr*∆ vs. wild type *S. pombe*. This result and its apparent discrepancy with our *mrps16* qPCR data could be explained by a) a real biological effect in which the splicing of *mrps16* circular RNA does not respond to increases in lariat abundance through inhibiting lariat debranching (e.g., the spliceosome is deposited co-transcriptionally, making a subset of lariats competent to splice; thus, preventing lariat degradation will not lead to increases in this pool) or b) technical differences between qPCR and RNA-Seq. In light of these findings, we believe that on average, there may be an increase in the *mrps16* circle:linear ratio in the debraching mutant; however, the experimental noise of both the RNA-seq and qPCR preclude us from making conclusions that belong in the Results section. Instead, we now comment on the results presented in [3] in our Discussion.

*Given the nature of the PCR assay it is important that the identity of each product is confirmed by sequencing. This has been done in some cases, but are all the band identities validated in this way? The bands in*
Figure 5
*are dismissed due to incorrect size, but what are these products? In several figures, bands are doublets (e.g.*
Figure 2*, lane 12), but there is no mention in the text of whether two different products were identified by sequencing. This could have implications for the mechanism and should be addressed at least for the double lariat*.

We have confirmed all major products by sequencing, excepting the exon-containing lariats in Figure 4 (previously, Figure 5), which were predicted to exist by [2] and are of the predicted size. Because RT across a 2’-5’ linkage introduces single nucleotide deletions and mutations, sequencing these products would require extensive cloning efforts. We have clarified our manuscript to make clear that our identification of the products in Figure 4 were based on size rather than sequence.

As you point out we observe a number of ‘shadow’ bands or ‘doublets,’ especially on our gel in Figure 2. In order to understand what these bands could be, we cloned the products in Figure 2, Lane 12 (the double lariat). We chose this product for two main reasons: 1. As you suggest, it may contain other products relevant to the mechanism. 2. The doublets are approximately equal intensity in this product, so it should be straightforward to see the distinct products by cloning (i.e., they should appear in approximately a 1:1 ratio).

We randomly selected 16 clones and performed a colony PCR and sequenced these clones (Figure 2—figure supplement 2). 10/16 clones were of the appropriate size to arise from this doublet. However, sequencing reveals that these products only differ by at most 2 nt’s in size, and that the products do not appear to represent distinct splice isoforms (see ClustalW2 alignment, Figure 2—figure supplement 3). We hypothesize that the reason for the doublet could be that, after many cycles of PCR, if the primer concentrations are uneven and limiting, a single-stranded product may form along with the double-stranded product; this single-stranded product may have a slight anomalous migration on the gel, leading to the doublet appearance we observe.

We did a literature search to determine if this problem has been addressed in the literature. While others have clearly encountered this problem, we could not find a scholarly discussion (rather, it is addressed on internet forums, for example: http://www.researchgate.net/post/Can_anyone_explain_the_secondary_DNA_bands_in_agarose_gel_besides_a-specific_amplification_in_PCR).

*The correlation of circle formation with deletion size within exon 2 seems very rough at best (Figure 6). Moreover, there are other possible effects of these mutations that do not seem to have been explored. First, it is not clear that circularization should vary continuously with length. Perhaps there is a minimum size below which an exon cannot be circularized. In fact looking at the plot, there might be such a discontinuity as their deletion reaches 100 bp. The authors also do not control for other effects that these deletions might have. One could imagine that a deletion simply removes a specific sequence required by the splicing machinery to successfully back-splice. This possibility might be controlled for by adding back different sequences to restore the length while maintaining the loss of elements specific to exon 2. Conversely, a deletion might eliminate exon skipping altogether. If the mutant exon 2 is very efficiently spliced, there may be no substrate for back splicing. This also could be examined*.

Thank you for raising this very interesting point. We agree that circularization efficiency should not vary continuously with length (we have removed our fitted curve so as not to imply this), but our data supports a general monotonic trend that larger deletions in *mrps16* exon 2 are less efficiently circularized. Both our analysis of endogenous circular RNA expression in other organisms (such as human, see [27]) and mutations in *mrps16* support the idea that exons smaller than 100nt can circularize. For example, the ∆90 nt (total exon length=91 nt) *mrps16* exon 2 isoform we analyzed by clonal analysis does form a circle, although at a much lower efficiency. Also, to address your last point, for this clone, we have analyzed the abundance of the lariat precursor production as well as splicing of the linear products exon1–exon 2 and exon 2–exon 3 by qPCR, which we now discuss in the manuscript.

Finally, for our *mrps16* mutants, you note that we did not control for other effects these deletions might have, such as removing a specific sequence required for backsplicing. We have clarified this point in the manuscript: we have indeed considered the possibility that a simple sequence motif has a significant impact on circular RNA abundance. This model would suggest that all deletions which contain deletions of a particular sequence motif (the one facilitating backsplicing) would result in attenuation of circular RNA abundance. We have specifically tested this model, and found it failed to fit the data. We included a figure that demonstrates this (Figure 5—figure supplement 5). We have also considered impacts of sequence deletions on structural properties of the *mrps16* circular RNA, discussed in the last section of the paper. However, we believe that computational structure predictions are not accurate enough to allow us to draw firm conclusions about this analysis, and have chosen not to emphasize it in the draft.

To reiterate, thank you again for raising these key questions: we have added discussion of these points in the manuscript which we believe have improved its clarity.

*Although the authors do a careful and thorough job of laying out their arguments, the paper is not an easy read. A number of choices of terminology and presentation are not standard usage and will confuse readers. These must be fixed both to adhere to the actual biochemical relationships of the molecules, and to avoid confusion with other previously established uses for the same terms. Major examples are listed here with others in the minor comments*.

*Figures*:

*In*
Figure 2*, the brackets showing the rolling circle RT-RCR products in lane 7 and 8 make lane 9 unreadable. Even though there is no signal in that lane, the reader should be able to see lane 9. Asterisks next to the bands in lanes 7 and 8 should work fine*.

Thank you for making this suggestion; we have indicated the rolling-circle products in 2A with ‘‡’.

Figure 2
*has a mirror image presentation of one sequence. This is confusing to assess and violates conventions against using mirror image letters and reading sequence 5' to 3'. Same for*
Figure 3—figure supplement 3*. One might justify not following these conventions if it clarified some aspect of the data, but in this case it would be clearer to follow them*.

Thank you for making this suggestion; we have changed the manuscript according to your suggestion.

*In*
Figure 5*, it seems like the red dashed lines are covering the place in the gel where the reader is supposed to assess the absence of signal. This is not possible with the dashed lines. An asterisk next to the lane would suffice to mark the position while still allowing the reader to see that there is no band there*.

Thank you for making this suggestion; we have replaced the dashes with a small asterisk.

*Terminology*:

*In the splicing field it is standard usage that the lariat intron and the spliced exons are both products of splicing. Because there are two catalytic steps, there are also intermediates called the free 5' exon and the* “*lariat intermediate*”*, which is the lariat intron still joined to exon 2. The use of the term* “*lariat intermediate*” *throughout this paper is confusing on two counts. First, the term has a different meaning from standard usage in the field. Second, the authors work quite hard to establish this exon-skipped lariat product as the precursor molecule for exon circularization. Possible alternative language for heading of the second section in the Results:* “*Quantitative analysis of splice isoforms demonstrates that the dominant pathway for circular RNA biogenesis in* mrps16 *uses the lariat product of exon skipping as a precursor in a second round of splicing*”*.*

*Finally what is a* “*terminated fork*”*? The direct backsplicing mechanism would involve a Y-shaped intermediate, similar to that produced by trans-splicing, instead of a lariat intermediate, and in the second step would produce a circle and a Y-shaped product instead of a lariat product*.

Thank you for pointing out our use of language that is potentially confusing. We have clarified it using alternative language according to your suggestions. The use of the terminology “terminated fork” was introduced in [17], and we adopted it for consistency. But as you point out, the splicing field has nomenclature for this product that precedes the use of the name “terminated fork,” and we have amended the manuscript to reflect this.

Minor points:

*1) In Figure 6–figure supplement 4, the authors show a secondary structure predicted for the exon. It would be interesting to know more about this. Do finer grained mutations that disrupt it affect circularization? As the authors state, a secondary structure would shorten the effective distance between the exon ends, but it may also possibly interfere with assembly of spliceosomal components or alter their bound conformation, which perhaps could positively affect the ability to back-splice. It would be helpful to resolve more clearly if the structure affected circle formation or not*.

Thank you for raising this question. We unfortunately are limited in the number and size of the mutations that we observe in this particular region and unfortunately small deletions are not highly represented here. Based on the data we currently have, we believe this structure might somehow repress circularization, perhaps by restricting the flexibility of the exon.

2) Is the ∆90 mutation plotted on Figure 6D?

No, it is not. We have now clarified this in the text.

*3) It may be the exposure of the gel, but in*
Figure 3—figure supplement 3
*the bands in B don't seem to agree with the bar graph in C. It appears that there is more of the BP-mutated one-exon circle relative to wild type than of the BP-mutated two-exon circle relative to wild type. The bar graph shows the opposite*.

This is an interesting point, and we realize that it would be impossible to know why unless you were acquainted with the raw data.

The reason for this is that the exon3 circle is more abundant (qPCR Ct= 27) than the intron-retained circle (qPCR Ct= 32). Thus, after 35 cycles (the # of cycles of PCR prior to running on the gel), the PCR for the exon 3 circle is close to saturation in both wt and ∆BP *pub1*, whereas a more dramatic difference can be observed for the intron-retained circle. We have illustrated this effect in the Figure 6.

Author response image 1.**DOI:**
http://dx.doi.org/10.7554/eLife.07540.026

4) In Figure 3—figure supplement 3, what do the two shades of gray mean in the bar graph?

One shade of grey represents products derived from the *wt pub1* plasmid while the other represents products derived from the ∆BP *pub1* plasmid. We apologize if this was not clear in our initial submission; we have created a legend to clarify this figure.